# Insecticidal Activities of *Atriplex halimus* L., *Salvia rosmarinus* Spenn. and *Cuminum cyminum* L. against *Dactylopius opuntiae* (Cockerell) under Laboratory and Greenhouse Conditions

**DOI:** 10.3390/insects13100930

**Published:** 2022-10-13

**Authors:** Imane Naboulsi, Karim El Fakhouri, Rachid Lamzira, Chaimae Ramdani, Gabin Thierry M. Bitchagno, Rachid Boulamtat, Widad Ben Bakrim, Ismail Mahdi, Aziz Aboulmouhajir, Abdelaziz Yasri, Mustapha El Bouhssini, Jane L. Ward, Mansour Sobeh

**Affiliations:** 1AgroBioSciences Program, Mohammed VI Polytechnic University, Lot 660, Hay Moulay Rachid, Ben Guerir 43150, Morocco; 2Organic Synthesis, Extraction and Valorization Laboratory, Faculty of Sciences Ain Chock, Hassan II University, Km 8 El Jadida Road, Casablanca 20000, Morocco; 3Entomology Laboratory, International Center for Agricultural Research in the Dry Areas (ICARDA), Rabat Institutes, Rabat 10100, Morocco; 4Computational and Analytical Sciences, Rothamsted Research, West Common, Harpenden AL5 2JQ, UK; 5African Sustainable Agriculture Research Institute (ASARI), Mohammed VI Polytechnic University (UM6P), Laayoune 70000, Morocco

**Keywords:** *Dactylopius opuntiae*, *Opuntia ficus-indica* L., bioinsecticides, *Atriplex halimus*, saponins

## Abstract

**Simple Summary:**

The wild scale insect *Dactylopius opuntiae* is one of the major insect pests of prickly pear *Opuntia ficus-indica* (L.) in Morocco. The present study investigated the insecticidal potential of the aqueous and hydroalcoholic extracts of *Atriplex halimus* (leaves), *Salvia rosmarinus* (leaves), and *Cuminum cyminum* (seeds) to control nymphs and adult females of *D. opuntiae* under laboratory and greenhouse conditions. Among the tested samples, the aqueous extract of *A. halimus* showed the highest activity on nymphs and adult females of *D. opuntiae* in the laboratory and when combined with black soap under greenhouse conditions. The difference in toxicity of the plant species in the study was correlated with their saponin content. These results provide evidence that the aqueous extract of *A. halimus* leaves could be incorporated into the management of the wild scale insect as an alternative to chemical insecticides.

**Abstract:**

The wild cochineal *Dactylopius opuntiae* (Hemiptera: Dactylopiidae) is one of the major insect pests of the prickly pear *Opuntia ficus-indica* (L.) in Morocco, a well-known fruit and vegetable crop of arid and semi-arid regions around the world. The present study investigated the insecticidal potential of six extracts (three aqueous and three hydroalcoholic (MeOH/H_2_O, 20/80 (*v*/*v*)) from *Atriplex halimus* (leaves), *Salvia rosmarinus* (leaves) and *Cuminum cyminum* (seeds) to control nymphs and adult females of *D. opuntiae* under laboratory and greenhouse conditions. Out of the tested samples, *A. halimus* aqueous extract showed the highest activity, inducing mortality rates of 67.04% (after 4 days) and 85% (after 8 days) on nymphs and adult females of *D. opuntiae*, respectively, at a concentration of 5% under laboratory conditions. It also showed the highest mortality rate of nymphs with 100% (4 days after application) and 83.75% of adult females (7 days after the second application) at a concentration of 5% when combined with black soap at 10 g/L under greenhouse conditions. The difference in the toxicity of plant species of the study was correlated with their saponin content. A total of 36 of these triterpene glucosides were suggested after a comprehensive LC-MS^n^ profiling of the most active extract, *A. halimus*, in addition to phytoecdysones and glycosylated phenolic acids and flavonoids. These findings provided evidence that the aqueous leaf extract of *A. halimus* could be incorporated in the management of the wild cochineal as an alternative to chemical insecticides.

## 1. Introduction

The Cactus pear, *Opuntia ficus-indica* (L.) Mill. (Caryophyllales: Cactaceae), is native to Mexico’s arid and semi-arid regions [1]. It was introduced to Morocco in 1770 as a good crop for years of low levels of rainfall, and for its multiple uses and applications [2]. The Cactus pear is widely represented in the Moroccan rural landscape because it requires low maintenance, and it is a natural boundary of crop fields. The cactus has experienced a resurgence of interest in several countries during the last decade because of its ecological, environmental, and socio-economic role. In most Mediterranean countries, cactus cultivation has become an intensive crop [3]. In Morocco, prickly pear plantations have been strongly encouraged by the Green Morocco Plan strategy (PMV-2008) as an alternative crop in less economically and culturally advantaged regions. The area of cacti has increased from 45,000 ha in the early 1990s to more than 150,000 ha in 2017 [4]. Several companies and cooperatives have been created to process the prickly pear, producing various products and by-products for different industrial, medicinal, pharmaceutical, and cosmetic sectors. This crop has suffered from the attack of the wild cochineal *Dactylopius opuntiae* (Cockerell) (Hemiptera: Dactylopiidae) in many countries, especially in the Mediterranean basin (Spain, Portugal, Morocco, Tunisia, Egypt, Israel, Lebanon, Turkey, and Greece), where it has become a serious pest of prickly pear crops [5]. This wild cochineal was first detected in the Sidi Bennour area, 70 km from El Jadida (western central Morocco) in September 2014 [6]. Due to the Mediterranean climate, *D. opuntiae* has spread rapidly, causing severe damage in most of the cactus-growing areas of Morocco, where farmers and cooperatives lost thousands of hectares of cactus, causing substantial socio-economic and environmental losses [6,7]. The females of this insect develop a white waxy layer that protects them from predators and parasitoids and provides them with perfect humidity; when they are crushed, they produce carminic acid, a natural dye used in the food, textile, and cosmetic industries [8]. Nymphs and adult females of *D. opuntiae* feed on plants by sucking sap from the cladode, leading to the destruction of the plant, and eventually killing it in the case of severe infestation.

In Morocco, significant progress has been made in developing integrated pest management (IPM) options to control this devastating insect [4]. Morocco’s National Office of Food Safety (ONSSA) has approved several pesticides to protect cactus crops and limit new infestations, including chlorpyrifos, pyriproxyfen, and mineral oils [9]. In addition to the resistance problem, these insecticides can severely affect human health and contaminate the environment, mainly plants and soil, and negatively impact natural enemies [6]. Therefore, the use of alternative approaches to control this pest (host plant resistance, natural enemies, plant extracts, vegetable and essential oils, and mycoinsecticides) has become a priority in the research field, as it is economically feasible and environmentally friendly [2].

Different botanical extracts are used as eco-friendly insecticides to control and manage different crop pests and plant diseases, due to the secondary compounds they contain, which have insecticidal properties against different insect pests [10]. In this context, Vigueras et al., (2009) [11] evaluated the efficacy of different plant extracts against *D. opuntiae* under laboratory conditions. They reported a high insecticidal effect of *Chenopodium ambrosioides* L., *Mentha piperita L, Mentha viridis* L., *Tagetes erecta* L., and *Tagetes florida* L. against second instars of *D. opuntiae* with 82, 92, 95, 98, and 99% of mortality at 72 h after treatments, respectively. Recently, Ramdani et al., (2021) [4] found that *Capsicum*
*annuum* L. fruit extract at 200 g/L in combination with black soap at 60 g/L could be used as one IPM component to control *D**. opuntiae*. Goncalves Diniz et al., (2020) [12] evaluated the effectiveness of the combinations of *Fusarium catingaense* L. isolates with extracts of *Nicotiana tabacum* L. and the mortality of the wild cochineal, and recorded 98.7% mortality in the greenhouse using 10% of aqueous extract of *N. tabacum* (*w*/*v*).

In this study, we selected three aromatic and medicinal plants, namely *Atriplex halimus* L. (leaves), *Salvia rosmarinus* Spenn. (leaves), and *Cuminum cyminum* L. (seeds), that grow or are cultivated in the region of Rhamna for their socio-economic importance for the habitats and their richness in different types and classes of secondary metabolites that can have a toxic impact on insect pests and scale insects. The major phytochemicals in *R. officinalis* and *C. cyminum* are phenolic metabolites, which are categorized into phenolic acids, flavonoids, and non-flavonoids [13,14,15,16], which include the major acids that characterize *S. rosmarinus* (syn. *R. officinalis*), rosmarinic, chlorogenic, and caffeic acids [13,14] and those of *C. cyminum*, chlorogenic, neochlorogenic, salicylic, cinnamic, and coumaric acids and coumarins [15,16]. However, the presence of these constituents depends on environmental conditions and extraction methods [17]. Likewise, in flavonoids, phenolic acids, and saponins, especially syringetin derivatives and atriplexoside A and B are the main constituent of *A. halimus* [18,19,20]. In this context, the objective of this work was to evaluate the toxicity effect of the selected botanical extracts applied alone or in combination with detergent as alternative biopesticides to control nymphs and adult females of *D. opuntiae* under laboratory and greenhouse conditions.

## 2. Material and Methods

### 2.1. Collection and Preparation of Plant Extracts

*Atriplex halimus* (Leaves), *S. rosmarinus* Spenn (leaves), and *C. cyminum* (seeds) were collected in Benguerir-Rhamna, Morocco (32°14′34.82″ N, 7°57′34.54″ W, Altitude 449 m). The materials were sorted, dried in a well-ventilated and dark place at room temperature, and ground using a laboratory mill (Retsch GM 200). The ground material was extracted by maceration at 50 °C for 30 min in 20/80 (*v*/*v*) MeOH/H_2_O and 100% water. Next, the extractive solutions were filtered twice on Whatman filter sheets, evaporated under reduced pressure till dryness and then lyophilized (Labconco FreeZone Legacy 2.5 L) to afford respective crude extracts. The extraction yields were 26.94, 14.89, and 15.92% (for 100 g of dry plant) for the aqueous extracts and 24.58, 10.12, and 11.36% (for 100 g of dry plant) for the 20/80 (*v*/*v*) MeOH/H_2_O extracts of *A. halimus, S. rosmarinus*, and *C. cyminum*, respectively.

### 2.2. Determination of Secondary Metabolites

#### 2.2.1. Total Polyphenol

The total phenol contents (TPC) of the extracts were estimated following the Folin–Ciocalteu (FC) method with slight modification [21,22]. A total of 200 µL of each extract (1 mg/mL) was mixed with 1 mL of FC reagent (Fresh diluted 1:10 with distilled water), allowed to react for 5 min at room temperature in the dark, and then 800 µL of 7.5% (*w*/*v*) of (Na_2_CO_3_) was added. After 30 min standing in the dark, the absorbance was measured at 760 nm against a blank. TPC were calculated based on the calibration curve of gallic acid and expressed as gallic acid equivalents (GAE) in micrograms per gram of the dried extract.

#### 2.2.2. Total Flavonoids

The total flavonoid contents (TFC) of extracts were determined using the aluminum chloride (AlCl_3_) method [23]. A total of 1 mL of extract at 1 mg/mL in methanol was added to 0.5 mL of AlCl_3_ (1.2%) and 0.5 mL of potassium acetate (120 mM) and allowed to react for 30 min at room temperature. The absorbance of each reaction mixture was measured at 415 nm to express the TFC as micrograms of quercetin equivalent per g of dry weight extract.

#### 2.2.3. Total Saponins

Saponins analysis was performed according to the method applied by Irigoyen et al., (2018) [24] by adding 250 µL of the tested extract (2.5 mg/mL) to 875 µL of Lieberman–Buchard reagent (16.7% acetic acid in 83.3% concentrated sulfuric acid). The absorbance of the vortexed solution was determined at 528 min after 30 min. The total saponin content of the extracts was evaluated as mg of Quillaija saponin per g of dry extract.

#### 2.2.4. UHPLC-MS Characterizations

The chemical profiling of extracts was conducted at Rothamsted Research (Harpenden, UK) on an Ultra-High-Performance Liquid Chromatography-Mass Spectrometry (UHPLC-MS) system equipped with a Dionex UltiMate 3000 RS UHPLC system coupled to a LTQ-Orbitrap Elite mass spectrometer (Thermo Fisher Scientific, Bremen, Germany). Samples were injected (10 μL) onto a reversed-phase Hypersil GOLD C18 selectivity HPLC column (3 μm, 30 × 2.1 mm i.d. Thermo Fisher Scientific) maintained at 35 °C. The elution consisted of a gradient of water/0.1% formic acid (A) and acetonitrile/0.1% formic acid (B) (0 min, 5% B; 0.1–20 min, 5–100% B; 20–25 min, 100% B; 25–26 min, 100–5% B; 26–31 min, 5% B) using a flow rate of 0.3 mL/min. Mass spectra were collected using an LTQ Orbitrap Elite with a heated ESI source (Thermo Scientific, Bremen, Germany), acquired in negative mode with a resolution of 120,000 over *m/z* 50–1500. The source voltage, sheath gas, auxiliary gas, sweep gas and capillary temperature were set to 2.5 kV, 35 (arbitrary units), 10 (arbitrary units), 0.0 (arbitrary units), and 350 °C, respectively. Default values were used for other acquisition parameters. Automatic MS–MS fragmentation was performed on the top four ions using an isolation width of *m/z*. Ions were fragmented using high-energy C-trap dissociation with a normalized collision energy of 65 and an activation time of 0.1 ms. Data were collected and inspected using Xcalibur v. 2.2 (Thermo Fisher Scientific). UV spectra were derived from PDA in LC-MS analyzes.

### 2.3. Insect Rearing

Healthy young cladodes of prickly pear cactus were planted in plastic pots (27 cm in diameter by 24 cm in height) filled with a mixture of soil, sand, and peat at greenhouse temperature of 30 ± 2 °C. They were exposed to heavily infested cladodes with *D. opuntiae* collected from Marchouch region (33°56′10″ N 6°69′21″ W). Each infested cladode was placed between two pots of healthy ones for 20 days to allow the development of mature pest females, which were carefully selected for further bioassays.

### 2.4. Laboratory Bioassays

The toxicity of plant extracts against nymphs and adult females of *D. opuntiae* was assessed at a temperature of 26 ± 2 °C, a relative humidity of 75% and a photoperiod of 14/10 h of light/darkness. A preliminary test was performed for all extracts at 5% against adult females of *D. opuntiae* by direct contact application to select the most effective extracts. Further bioassays were conducted at four concentrations (0.625, 1.25, 2.5, and 5%) of each plant extract using a 1 L hand sprayer (0.5 mL of the tested extract per petri dish) in a completely randomized design with five replications per concentration for each treatment. Ten females of the same age and ten first instar nymphs of *D. opuntiae* were deposited on pieces of cladodes of the same size with an entomological brush and placed in glass Petri dishes (90 mm × 15 mm). Control nymphs and female cochineal were treated with water. The mortality of adult females was evaluated every 24 h within the eight days of treatments using a binocular microscope (Motic DM-143), whereas nymphs’ mortality was recorded 1, 3, 24, 48, 72, and 96 h after exposure. Dead females lost their natural color to dark brown and desiccation of their bodies, while dead nymphs showed no movement and severe color changes.

Mortality was calculated according to Abbott’s formula [25]:$$\mathrm{Corrected}\;\mathrm{Mortality}\;\left(\%\right)=\frac{\left[\%\mathrm{mortality}\;\mathrm{in}\;\mathrm{treatment}-\%\mathrm{mortality}\;\mathrm{in}\;\mathrm{control}\right]}{\left[100-\%\mathrm{mortality}\;\mathrm{in}\;\mathrm{control}\right]}\times 100$$

### 2.5. Greenhouse Trials

Healthy young cladodes of *O. ficus*-*indica* were planted in pots (27 cm in diameter by 24 cm high) filled with a mixture of soil, sand, and peat at a temperature of 30 ± 2 °C. The cladodes were infested artificially using the method described above. Treatments were applied to selected cladodes at medium infestation levels (26–50%) using a modified rating scale from Silva (1991). In general, the approximate number of colonies ranged from 41 to 80 per cladode. The ratings for the infestation level were performed using a scale of 1–6 using: (1) 1 to 5 colonies, (2) 6 to 15 colonies, (3) 16 colonies up to 25% of the cladode covered by cochineal, (4) 26 to 50%, (5) 51 to 75%, (6) 76 to 100% of the cladode covered by cochineal.

Experiments were conducted in a randomized complete block design with four replications. Cladodes were sprayed beforehand with black soap at 10 g/L, which served to remove the cuticular wax and exposed the females and nymphs to tested plant extracts, and then the 5% treatments were applied twice using a 1 L hand sprayer (5 mL per cladode) at a seven day interval [4]. Cladodes treated with water alone and water mixed with black soap served as controls. Nymphs’ mortality was recorded at 1, 2, 3, and 4 days after the treatment and females’ mortality was assessed 3, 5, and 7 days after the first and second application, each.

### 2.6. Data Analysis

Mortality percentages were transformed into angular values (arcsine √P) before the statistical analysis. Under laboratory conditions, the transformed percentages were subjected to two-way analysis of variance (ANOVA) (concentration and extract source). Under greenhouse conditions, the transformed percentages were subjected to one-way analysis of variance. Lethal concentration for 50% and 90% mortality (LC50 and LC90, respectively) were determined using probit analysis [26]. The means were compared by Newman–Keuls tests at *p* < 0.05 using XLSTAT 2021.1 by Addinsoft.

## 3. Results

### 3.1. Laboratory Bioassays

#### 3.1.1. Insecticidal Effects of Different Plant Extracts on *D. opuntiae*: Preliminary Bioassays

Initially, we evaluated the insecticidal activities of the six extracts at a concentration of 5% against adult females of *D. opuntiae* using foliar application under lab conditions. The results revealed a significant difference in mortality of females between the six tested extracts after 5 days of application, where the aqueous extracts were more effective than the hydroalcoholic extracts irrespective of the plant species; see Table 1. The corrected mortality rates significantly increased to 89.28% for both *A. halimus* aqueous and hydroalcoholic extracts 8 days after application (DF = 6, *p* < 0.0001). However, *S. rosmarinus* and *C. cyminum* extracts possessed the lowest insecticidal activity against *D. opuntiae* at 5%, with a corrected mortality rate not exceeding 17.85%.

#### 3.1.2. Effect of *A. halimus* Extracts on *D. opuntiae* Nymphs and Adult Females

Based on the preliminary results, the insecticidal activities of the most toxic extracts (*A. halimus*, aqueous and hydroalcoholic) against the nymphs and adult females were tested using four different concentrations; see Table 2 and Table 3. The results showed a significant difference in mortality of nymphs from 24 h after treatment in a dose- and time-dependent manner; see Table 2. The ANOVA analysis showed a highly significant difference (DF = 2, *p* < 0.0001) in mortality of *D. opuntiae* nymphs 4 days after the spray. The methanolic extract of *A. halimus* leaves and the aqueous extract at 5% concentration recorded the highest corrected mortality rates with 77.27% and 67.04%, respectively.

Data analysis showed a highly significant difference in the insecticidal effect of both *A. halimus* extracts on *D. opuntiae* females at different exposure times (*p* < 0.001) (Table 3).

The mortality of adult females increased significantly after 5 days of application (DF = 3, *p* < 0.0001). The increasing concentrations of the extracts significantly increased the mortality of females for different periods. At 8 days after treatment, the highest corrected mortality, 85.00% of females, was recorded for the aqueous extract of *A. halimus* at 5% concentration followed by hydromethanolic extract with 52.50% mortality for the same concentration (DF = 3, *p* < 0.0001).

Lethal concentrations LC_50_ and LC_90_ varied between the tested plant extracts. The lethal concentration values of plant extracts tested against *D. opuntiae* nymphs showed that *A. halimus* aqueous extract recorded LC_50_ = 3.25% and LC_90_ = 5.47%, and *A. halimus* extract 20% methanol recorded LC_50_ = 3.06% and LC_90_ = 5.20% (Table 4) four days after treatment. The lethal concentration values of plant extracts tested against *D. opuntiae* females indicated that the *A. halimus* aqueous extract (LC_50_ = 2.99% and LC_90_ = 5.29%) were more effective than *A. halimus* extract 20% methanol (LC_50_ = 3.43% and LC_90_ = 7.13%) seven days after treatment (Table 5).

### 3.2. Greenhouse Trials

According to the results mentioned above, we further evaluated the insecticidal effects of *A. halimus* aqueous extract against *D. opuntiae* nymphs and adult females under greenhouse conditions. No evidence of phytotoxicity was observed (chlorosis, change in coloration, and necrosis lesions) in treated cladodes during and after treatment. The statistical analysis showed significant differences in mortality of adult females and nymphs caused by the aqueous extract of *A. halimus* (5%) in combination with black soap during the different exposure periods (Table 6 and Table 7). At day 2 after treatment, *A. halimus* with the detergent induced high mortality of nymphs with 90.00%, which increased significantly up to 100% at day 4 after application (DF = 2, *p* < 0.0001) (Table 6).

The ANOVA showed significant differences in mortality of adult females caused by different treatments and the control starting from day 7 after the first spray (Table 7). The mortality of adult females remained low and did not exceed 40% during the first week after application. The second application significantly increased the mortality rate of the females from 63.75% (day 10 after spraying) to 83.75%, (day 14 after application) (DF = 2, *p* < 0.0001), while the solution in water of black soap at 10 g/L used as a control induced a very low female mortality rate (3.50%) following the same treatment (Table 7).

### 3.3. Phytoconstituents Composition

#### 3.3.1. Total Polyphenol, Flavonoid, and Saponin Contents

To correlate the observed activities with the phytoconstituents of the tested samples, we initially quantified their total phenols, flavonoids and saponins contents; see Table 8. Overall, aqueous extracts were shown to contain higher levels of phenolic compounds compared to hydroalcoholic extracts. Water extracts of *S. rosmarinus* featured the highest total polyphenol (170.03 ± 12.33 mg GAE/g DW) and flavonoid (33.46 ± 1.46 mg QE/g DW) contents compared to *C. cyminum* and *A. halimus*. However, both aerial part extracts of *A. halimus* showed high contents of saponins (24.09 ± 0.71 mg SSE/g DW) compared to the other plant extracts, which explains the foamy aspect of the extracts.

#### 3.3.2. Phytochemical Profiling of *A. halimus* Extract

Annotation of the secondary metabolites of the most active aqueous extract, *A. halimus*, was carried out using LC-MS/MS (Figure 1). Inspection of the total ion chromatogram allowed a list of major components to be generated (Table 9) and use of the MSMS data suggested the identities of the observed metabolites. A total of 58 compounds were detected according to the following criteria: (i) the calculated mass was close to the experimental mass (error ≤ 5 ppm); (ii) fragmentation patterns (MS/MS) were in agreement with the proposed structure; (iii) Putative compounds belonged to a class of compounds previously listed to occur in the studied plant species (or genus).

Most of them were triterpenoid saponins and hydroxylated and glycosylated compounds derived mainly from olean-12-en-28-oic acid or oleanolic acid (37 compounds), in addition to phytoecdysones (e.g., Dimorphamide A derivative). Simple organic acids such as citric acid, succinic acid, hydroxymethylglutaric acid, and phenolic acids including protocatechuic acid were also identified. Flavonoid aglycones and sulphated derivatives, including isorhamnetin and isorhamnetin sulfate, atriplexoside A, narcissin, and quercetin derivatives were also detected.

Under the collision energies employed, the fragmentation pattern of the saponins showed the molecular ion followed by low intensities of ions resulting from the successive loss of sugar moieties starting from the external unit. A loss of *m/z* 162 was characteristic of an ether glucosidation (rather than an ester glucosidation which releases predominantly *m/z* 180). Glucuronic acid ether-type conjugation was characterized by a loss of *m/z* 176. The mass spectrum also showed ions from the aglycone (*m/z* 455, 469, 471, 515, etc.) depending on the oxidation pattern of the molecule. As usual in olean-12-en-28 oic acid, ions from the retro-Diels–Alder (RDA) degradation mechanism were often evidenced with very low intensities at *m/z* 206 and 248 for the aglycone at *m/z* 455 or at *m/z* 207 and 262 (eventually *m/z* 221 and 248) for the aglycone at *m/z* 469. Flavonoids were also identified based on their fragmentation patterns displaying the characteristic aglycone ion in the MS^2^. Isorhamnetin was believed to constitute the aglycone of most of the flavonoids owing to characteristic peaks in the MS^2^ at *m/z* 152 and 163 from the RDA in ring C. The number of substitutions on the isorhamnetin backbone was determined to be identical to the number of protons lost from its corresponding mass (*m/z* 315) when comparing to the aglycone ion of a particular flavonoid (e.g., monodesmodic with aglycone at *m/z* 315 and bidesmodic with aglycone at *m/z* 314). Sugars were identified based on the losses of their corresponding masses (*m/z* 162 for glucoside, *m/z* 146 for rhamnose, *m/z* 132 for arabinose-like sugar). Finally, a compound was putatively annotated when it has been previously reported from either the studied plant or one of the species of the genus [18,20,27,28,29] and its MS characteristics matched with fragmentation patterns listed above. The other compounds were annotated by comparison of their respective MS characteristics with reported MS data in the literature [30,31,32,33].

## 4. Discussion

The present study demonstrated the insecticidal activity of the aqueous and hydroalcoholic extracts of *A. halimus* against *D. opuntiae*, quantified their total phenol, flavonoid and saponin contents, and annotated their secondary metabolites. Both extracts were able to kill nymphs and adult females under laboratory conditions, while only the aqueous extract preserved this activity at the greenhouse scale where the mortality rate exceeded 70%. The *A. halimus* aqueous extract showed substantial activity and was able to kill the nymphs and adult females by 100% and 83.75%, respectively, when tested at 5% in a combination with 10 g/L black soap. The observed activities might be attributed to the high contents of saponins that amounted to 24.09 mg SSE/g DW at *A. halimus* aqueous extract. These in vitro results were in line with the LC-MS studies in which we identified 58 secondary metabolites, and among them 37 suggested olean-12-en-28-oic or oleanolic acids derivatives.

*A. halimus* is a rich source of phenolic compounds, such as various flavonol glycosides isolated from 60% methanolic fraction, namely syringetin 3-*O*-beta-D-rutinoside, syringetin 3-*O*-beta-D-glucopyranoside, and isorhamnetin 3-*O*-beta-D-rutinoside or narcissin [20], which agree with our investigation. In addition, the investigation of bioactive chemical compounds conducted by Kabbash and Shoeib (2012) on aerial parts of the plant identified two flavonol glycosides, designated as atriplexoside A [3′-*O*-methylquercetin-4′-*O*-β-apiofuranoside-3-*O*-(6″-*O*-α-rhamnopyranosyl-β-glucospyranoside)] that was present in our extract and atriplexoside B [3′-*O*-methylquercetin-4′-*O*-(5″-*O*-β-xylopyranosyl-β-apiofuranoside)-3-*O*-(6″-*O*-α-rhamnopyranosyl-β-glucoside)], as well as two phenolic glycosides, an ecdysteroid, a megastigman, and two methoxylated flavonoid glycosides [18]. The present study’s results corroborate the findings of Donia et al., (2012) that also isolated hederagenin-3-*O*-β-D-glucuronopyranoside and oleanolic acid-3-*O*-β-D-glucuronopyranoside (calenduloside E) from *Atriplex farinosa* Forssk [34]. The oleanolic acid 3-*O*-β-D-glucopyranosyl-28-*O*-β-D-glucopyranoside was extracted by Kumarihamy et al., (2015) from the EtOH extract of *A. canescens* (Pursh) Nutt. leaves [35].

Plant secondary metabolites, such as alkaloids, tannins, saponins, lectins, terpenoids, phenolics, phenylpropanoids, and glycosides, have shown significant insecticidal activity against various crop pests including the wild cochineal, *D. opuntiae* [4,10,36,37,38]. Our findings are consistent with instances in the literature [39,40]. El-Gougary (1998) [39] has demonstrated the insecticidal activity of the ethanol extract of *A. halimus* against the cotton caterpillar *Spodoptera littoralis* (Boisduval) (Lepidoptera: Noctuidae) with an LD_50_ value of 5.6 mg/larva. The ethanolic extract and petroleum ether of *A. halimus* were also used in a ratio of 1:1 (*w/w*) with organo-phosphorus insecticide, pirimiphos-methyl (actellic) and chlorpyrifos-methyl (reldan), on the adult insect *Tribolium castaneum* (Herbst) (Coleoptera: Tenebrionidae). The petroleum-ether extracts strongly furnished synergistic effects with chlorpyrifos-ether, while the ethanol extracts displayed synergistic effects with pirimiphos-methyl with ratios of 2.8× and l.4×, respectively [39]. Kamal et al., 2017 [40] studied the insecticidal activities of saponins and various solvent extracts of *Atriplex laciniata* against various pests. Crude saponins from *Atriplex laciniata* L. were most effective against *Heterotermes indicola* (Wasmann) (Isoptera: Rhinotermitidae), *Rhyzopertha dominica* F. (Coleoptera: Bostrichidae), and *T. castaneum* (Herbst), causing 90.36 ± 0.6, 80.1 ± 0.7, and 86.5 ± 0.6% mortalities, respectively. The saponin extract also showed the highest activity against *Monomorium pharaonic* (Linnaeus) (Hymenoptera: Formicidae), with an LD_50_ of 78 mg/Ml.

Likewise, the low to moderate activity encountered for the two other plants of our study was consistent with a previous study by Idris et al., (2021) [41]. Indeed, the ethanolic extract (50%) of *S. rosmarinus* (syn. *R. officinalis*) leaves showed moderate toxicity on the adult females of the wild cochineal *D. opuntiae* after five days of application in both laboratory and field conditions. This mortality rate reached 38.7 ± 4.0% in laboratory conditions and 25.6 ± 5.1% in field treatments after 5 days of exposure. The wild cochineals were, however, sensitive to *Virginia tobacco* L. extract, which induced considerable toxicity on nymphs and adult females after 5 days of application under laboratory conditions with a mortality rate of 91.7 ± 5.1% and 83.4 ± 3.1%, respectively. In addition, the mortality rates were 87.1 ± 3.8% and 78.5 ± 3.6% for nymphs and adult females, respectively, after 5 days of treatment in the field [41].

The insecticidal activity of *C. cyminum* extracts has been rarely studied. However, in a recent study by Khorrami et al., (2019), it was revealed that the Fe₃O₄ methanolic extract of *C. cyminum* was significantly more repellent against the potato insect pest *Phthorimaea operculella* (Zeller) [42].

Interesting studies have been published in recent years regarding the insecticidal activities of many plant extracts against *D. opuntiae*. Lopes et al., (2018) showed that the extracts of leaves and pods of *Libidibia ferrea* (Mart. ex Tul.) L. P. Queiroz were more effective against the wild cochineal, causing 81% mortality of nymph II and 97% of adult females, ten days after treatment, while *Agave sisalana* (Perrine) extracts controlled only adult females, causing 51–97% mortality. According to this study, the insecticidal activity of *L. ferrea* extracts can be explained by the toxicity presented by their secondary compounds such as flavonoids, saponins, tannins, gallic acid coumarins, steroids, and phenolic compounds [37]. Vigueras et al. (2009) found similar results by assessing the extracts of *C. ambrosioides*, *M. piperita*, *M. viridis*, *T. erecta*, and *T. florida* Sweet on *D. opuntiae*, with mortality ranging from 35% of nymphs to 98% for adult females, suggesting that terpenoids in the extracts may be responsible for this toxicity [11].

On the other hand, Ramdani et al. (2021) reported that the application of *Capsicum annuum* at 200 g/L in combination with black soap at 60 g/L showed excellent control of *D. opuntiae* females with 87.31% at 7 days after application under field conditions. However, the use of the *C. annum* extract alone at 200 g/L showed less mortality of females with only 18.40% [43]. The current study showed that the insecticidal effect of plant extracts against *D. opuntiae* nymphs and females was enhanced by applying the detergent black soap. However, further studies are necessary to establish an adequate formulation based on aqueous *A. halimus* extract or its active chemical compounds, compatibility with other biopesticides, and impact on natural enemies. Laboratory and greenhouse studies should be supported by field trials to confirm the effectiveness of this bioproduct against wild cochineal and determine the most effective method of the application under field conditions.

## 5. Conclusions

The findings of the current study indicate that an aqueous extract of *A. halimus* possessed a good insecticidal activity against *D. opuntiae*. Nymphs and adult females of *D. opuntiae* were more sensitive to the aqueous extract of *A. halimus* leaves compared to the 20% methanolic extract from the same plant and compared to the other tested plant extracts. Phytochemical determination revealed a high concentration of total saponin. Additionally, LC-MS characterization indicated the presence of valuable biologically active compounds and a large number of saponin derivatives, especially triterpenoid saponins, which could be responsible for insect mortality. However, further studies are required on individual purified components to determine those most effective for wild cochineal mortality, and in addition, to develop formulations based on the bioactive molecules with detergent to dissolve to wax as new and more effective biopesticides.

## Figures and Tables

**Figure 1 insects-13-00930-f001:**
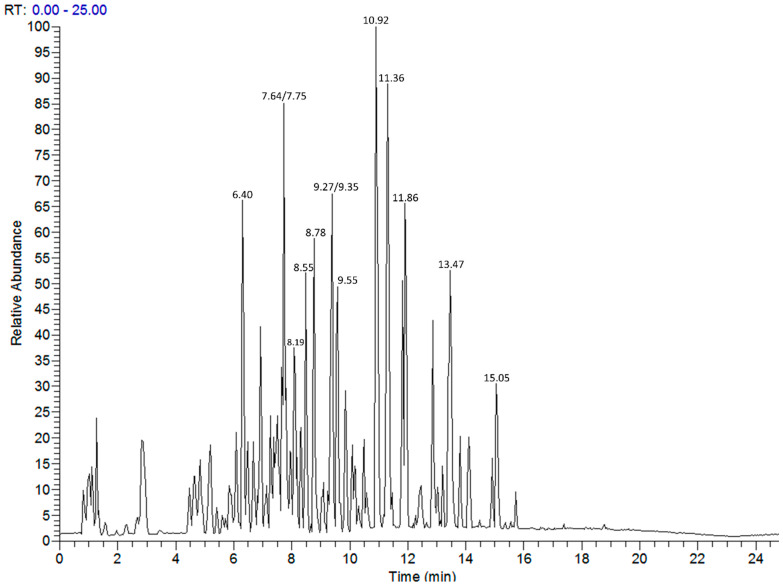
LC-MS profile of *A. halimus* aqueous extract. Data collected in negative ion mode following reversed phase separation. Numbers relate to retention time.

**Table 1 insects-13-00930-t001:** Mean percentage mortality ± SE of *D. opuntiae* adult females after exposure to six plant extracts at 5%.

Treatments	Mortality (%)							
	1 d	2 d	3 d	4 d	5 d	6 d	7 d	8 d
***A. halimus* aqueous extract**	6.67 ± 3.33	13.33 ± 3.33 ^a^	20.00 ± 5.77 ^a^	30.00 ± 11.55	40.00 ± 5.77 ^a^	53.33 ± 3.33 ^a^	60.00 ± 10.00 ^a^	90.00 ± 5.77 (89.29) ^a^
***A. halimus* hydroalcoholic extract**	0.00	0.00 ^b^	0.00 ^b^	13.33 ± 8.82	27.33 ± 6.36 ^ab^	34.33 ± 8.69 ^b^	55.67 ± 17.95 ^a^	90.00 ± 10.00 (89.29) ^a^
***C. cyminum* aqueous extract**	3.33 ± 3.33	3.33 ± 3.33 ^ab^	3.33 ± 3.33 ^b^	3.33 ± 3.33	3.33 ± 3.33 ^b^	3.33 ± 3.33 ^c^	10.00 ± 5.77 ^b^	16.67 ± 12.02 (10.71) ^b^
***C. cyminum* hydroalcoholic extract**	0.00	0.00 ^b^	0.00 ^b^	0.00	3.33 ± 3.33 ^b^	3.33 ± 3.33 ^c^	16.67 ± 8,82 ^b^	23.33 ± 12.02 (17.85) ^b^
***S. rosmarinus* aqueous extract**	0.00	7.67 ± 3.84 ^ab^	15.33 ± 3.33 ^a^	19.00 ± 7.00	22.67 ± 10.67 ^ab^	22.67 ± 10.67 ^bc^	22.67 ± 10.67 ^b^	22.67 ± 10.67 (17.14) ^b^
***S. rosmarinus* hydroalcoholic extract**	0.00	6.67 ± 3.33 ^ab^	10.00 ± 0.00 ^ab^	10.00 ± 0.00	10.00 ± 0.00 ^b^	10.00 ± 0.00 ^c^	13.33 ± 3.33 ^b^	16.67 ± 3.33 (10.71) ^b^
**Check (water)**	0.00	0.00 ^b^	0.00 ^b^	0.00	3.33 ± 5.77 ^b^	3.33 ± 5.77 ^c^	3.33 ± 5.77 ^b^	6.67 ± 5.77 ^b^
***p*-value**	0.109	0.019	0.000	0.036	0.002	0.000	0.005	<0.0001

Means in the same column followed by different letter(s) are significantly different based on Newman–Keuls test (*p* < 0.05). Percentages in parentheses represent the corrected mortality of the last day.

**Table 2 insects-13-00930-t002:** Mean percentage mortality ± SE of *D. opuntiae* nymphs after exposure to aqueous and methanolic extracts of *A. halimus*.

Treatments	Concentrations (mg/L)	Mortality (%)					
		1 h	3 h	1 d	2 d	3 d	4 d
***A. halimus* aqueous extract**	0.625	0.00 ^a^	4.00 ± 4.00	4.00 ± 4.00 ^b^	7.20 ± 3.14 ^b^	12.00 ± 7.35 ^bc^	14.00 ± 7.48 (2.27) ^b^
	1.25	0.00 ^a^	4.00 ± 4.00	6.00 ± 2.45 ^b^	10.00 ± 3.16 ^b^	16.00 ± 4.00 ^bc^	26.00 ± 6.00 (15.90) ^b^
	2.5	0.00 ^a^	5.20 ± 3.32	7.20 ± 3.14 ^b^	11.20 ± 3.38 ^b^	17.20 ± 2.52 ^bc^	35.00 ± 4.09 (26.14) ^b^
	5	2.00 ± 2.00 ^a^	5.20 ± 3.32	8.00 ± 2.00 ^b^	20.00 ± 5.48 ^b^	33.00 ± 7.68 ^b^	71.00 ± 8.43 (67.04) ^a^
***A. halimus* hydroalcoholic extract**	0.625	0.00 ^a^	4.00 ± 4.00	6.00 ± 4.00 ^b^	10.00 ± 5.48 ^b^	14.00 ± 5.10 ^bc^	22.00 ± 8.00 (11.36) ^b^
	1.25	0.00 ^a^	6.00 ± 4.00	8.00 ± 3.74 ^b^	10.00 ± 5.48 ^b^	16.00 ± 4.00 ^bc^	22.00 ± 5.83 (11.36) ^b^
	2.5	0.00 ^a^	8.00 ± 3.74	10.00 ± 3.16 ^b^	12.00 ± 4.90 ^b^	16.00 ± 6.78 ^bc^	34.00 ± 7.31 (25.00) ^b^
	5	2.00 ± 2.00 ^a^	14.00 ± 6.00	36.00 ± 15.68 ^a^	52.00 ± 13.56 ^a^	71.00 ± 11.45 ^a^	80.20 ± 8.43 (77.27) ^a^
**Check (Water)**		0.00 ^a^	0.00	0.00 ^b^	4.00 ± 2.45 ^b^	4.00 ± 2.45 ^c^	12.00 ± 2.00 ^b^
***p*-Value (Model)**		0.613	0.483	0.011	0.000	<0.0001	<0.0001
***p*-Value (Concentration × Treatments)**		0.530	0.702	0.092	0.035	0.009	0.702

Means in the same column followed by a different letter(s) are significantly different based on Newman–Keuls test (*p* < 0.05). Percentages in parentheses represent the corrected mortality of the last day.

**Table 3 insects-13-00930-t003:** Mean percentage mortality ± SE of *D. opuntiae* females after exposure to aqueous and methanolic extracts of *A. halimus*.

Treatments	Concentration (mg/L)	Mortality (%)					
		3 d	4 d	5 d	6 d	7 d	8 d
***A. halimus* aqueous extract**	0.625	6.00 ± 4.00 ^b^	8.00 ± 3.74 ^b^	16.00 ± 9.27 ^b^	16.00 ± 9.27 ^c^	20.00 ± 10.49 ^c^	22.00 ± 9.70 (2.50) ^cd^
	1.25	12.00 ± 3.74 ^ab^	18.00 ± 6.63 ^ab^	26.00 ± 4.00 ^b^	28.00 ± 3.74 ^bc^	34.00 ± 4.00 ^bc^	42.00 ± 5.83 (27.50) ^bcd^
	2.5	14.00 ± 5.10 ^ab^	18.00 ± 3.74 ^ab^	26.00 ± 2.45 ^b^	30.00 ± 7.07 ^bc^	34.00 ± 5.10 ^bc^	44.00 ± 6.78 (30.00) ^bcd^
	5	26.00 ± 6.78 ^a^	38.00 ± 5.83 ^a^	68.00 ± 9.70 ^a^	76.00 ± 6.78 ^a^	78.00 ± 6.63 ^a^	88.00 ± 5.83 (85.00) ^a^
***A. halimus* hydroalcoholic extract**	0.625	6.00 ± 4.00 ^b^	6.00 ± 4.00 ^b^	12.00 ± 4.90 ^b^	14.00 ± 5.10 ^c^	26.00 ± 10.77 ^bc^	26.00 ± 6.78 (7.50) ^cd^
	1.25	6.00 ± 2.45 ^b^	8.00 ± 2.00 ^b^	26.00 ± 12.08 ^b^	28.00 ± 11.14 ^bc^	32.00 ± 10.20 ^bc^	42.00 ± 7.35 (27.50) ^bcd^
	2.5	10.00 ± 3.16 ^ab^	20.00 ± 6.32 ^ab^	36.00 ± 8.12 ^b^	42.00 ± 4.90 ^bc^	44.00 ± 5.10 ^bc^	52.00 ± 5.83 (40.00) ^bc^
	5	10.00 ± 0.00 ^b^	32.00 ± 8.60 ^a^	44.00 ± 8.12 ^b^	52.00 ± 10.20 ^b^	58.00 ± 10.68 ^ab^	62.00 ± 8.60 (52.50) ^b^
**Check (Water)**		4.00 ± 2.45 ^b^	8.00 ± 4.90 ^b^	12.00 ± 8.00 ^b^	18.00 ± 8.00 ^c^	18.00 ± 8.00 ^c^	20.00 ± 7.07 ^d^
***p*-Value (Model)**		0.014	0.001	0.000	<0.0001	0.000	<0.0001
***p*-Value (Concentration × Extract)**		0.234	0.716	0.203	0.150	0.294	0.096

Means in the same column followed by a different letter(s) are significantly different based on Newman–Keuls test (*p* < 0.05). Percentages in parentheses represent the corrected mortality of the last day.

**Table 4 insects-13-00930-t004:** Lethal concentrations (LC_50_ and LC_90_) of *A. halimus* extract, against *D. opuntiae* nymphs (at *p* < 0.05) after exposure to different times.

Extract	Time	LC_50_	LC_90_	Coefficient of Determination	Graph Equation
***A. halimus* extract in water**	4 days	3.25	5.47	R^2^ = 0.8955	y = 18x − 8.5
	3 days	7.24	13.47	R^2^ = 0.8049	y = 6.42x + 3.5
	2 days	12.07	22.17	R^2^ = 0.8556	y = 3.96x + 2.2
	1 day	-	-	-	-
	3 h	-	-	-	-
***A. halimus* extract 20% methanol**	4 days	3.06	5.20	R^2^ = 0.7572	y = 18.66x − 7.1
	3 days	3.71	6.05	R^2^ = 0.6284	y = 17.1x − 13.5
	2 days	4.77	7.89	R^2^ = 0.638	y = 12.8x − 11
	1 day	4.57	8.91	R^2^ = 0.7101	y = 9.2x − 8
	3 h	-	-	-	-

**Table 5 insects-13-00930-t005:** Lethal concentrations (LC_50_ and LC_90_) of *A. halimus* extracts, against *D. opuntiae* females (at *p* < 0.05) after exposure to different times.

Extract	Days	LC_50_	LC_90_	Coefficient of Determination	Graph Equation
***A. halimus* extract in water**	8	2.54	4.58	R^2^ = 0.9728	y = 19.667x
	7	2.99	5.29	R^2^ = 0.7938	y = 17.4x − 2
	6	3.19	5.38	R^2^ = 0.7921	y = 18.2x − 8
	5	3.53	6.09	R^2^ = 0.7567	y = 15.6x − 5
	4	5.78	10.22	R^2^ = 0.8526	y = 9x − 2
	3	8.23	14.68	R^2^ = 0.9109	y = 6.2x − 1
***A. halimus* extract 20% methanol**	8	2.10	4.57	R^2^ = 0.9847	y = 11.8x + 16
	7	3.43	7.13	R^2^ = 0.972	y = 10.8x + 13
	6	3.75	6.88	R^2^ = 0.9942	y = 12.8x + 2
	5	4.43	8.21	R^2^ = 0.9839	y = 10.6x + 3
	4	6.22	10.67	R^2^ = 0.931	y = 9x − 6
	3	-	-	-	-

**Table 6 insects-13-00930-t006:** Mean percentage ± SE of *D. opuntiae* nymphs after exposure to *A. halimus* extract and its combination with black soap under greenhouse conditions.

Treatments/Exposure Time	Mortality (%)			
	1 d	2 d	3 d	4 d
***A. halimus* extract in water (5%) + Black soap (10 g/L)**	57.50 ± 7.50 ^a^	90.00 ± 5.77 ^a^	90.00 ± 5.77 ^a^	100.00 ± 0.00 ^a^
**Black soap + water**	14.00 ± 5.89 ^b^	21.25 ± 8.51 ^b^	35.00 ± 11.37 ^b^	41.25 ± 12.97 ^b^
**Water**	0.00 ^b^	0.00 ^c^	0.00 ^c^	0.00 ^c^
***p*-Value**	<0.0001	<0.0001	<0.0001	<0.0001

Means in the same column followed by a different letter(s) are significantly different based on Newman–Keuls test (*p* < 0.05).

**Table 7 insects-13-00930-t007:** Mean percentage ± SE of *D. opuntiae* females after exposure to *A. halimus* extract using two sprays.

Treatments/Exposure Time	% Mortality of Females after First Spray			% Mortality of Females after Second Spray		
	3 d	5 d	7 d	10 d	12 d	14 d
***A. halimus* extract in water (5%) + Black soap (10 g/L)**	15.75 ± 6.66 ^a^	27.50 ± 7.98 ^a^	42.50 ± 6.08 ^a^	63.75 ± 14.90 ^a^	68.00 ± 14.67 ^a^	83.75 ± 8.37 ^a^
**Black soap + water**	0.00 ^b^	0.00 ^b^	2.00 ± 0.82 ^b^	3.00 ± 1.73 ^b^	3.00 ± 1.73 ^b^	3.50 ± 2.22 ^b^
**Water**	0.00 ^b^	0.00 ^b^	0.00 ^b^	0.00 ^b^	0.00 ^b^	0.00 ^b^
***p*-Value**	0.012	0.001	<0.0001	0.000	<0.0001	<0.0001

Means in the same column followed by a different letter(s) are significantly different based on Newman–Keuls test (*p* < 0.05).

**Table 8 insects-13-00930-t008:** The total contents ± SE of polyphenols, flavonoids, and saponins in the tested extracts.

Assay	Plant Extracts					
	*C. cyminum*		*S. rosmarinus*		*A. halimus*	
	Aqueous	Hydroalcoholic	Aqueous	Hydroalcoholic	Aqueous	Hydroalcoholic
**TPC (mg GAE/g DW)**	45.51 ± 1.45	42.81 ± 3.52	170.03 ± 2.33	166.76 ± 4.95	36.29 ± 2.18	33.28 ± 2.06
**TFC (mg QE/g DW)**	22.63 ± 0.24	20.82 ± 0.28	33.46 ± 1.46	24.68 ± 1.02	8.11 ± 0.30	4.57 ± 0.45
**Saponines (mg SSE/g DW)**	2.73 ± 0.34	1.81 ± 0.47	4.36 ± 0.52	3.63 ± 0.31	24.09 ± 0.71	18.90 ± 1.15

DW: dry weight, QE: quercetin equivalent, SSE: standard saponin equivalent, GAE: gallic acid equivalent.

**Table 9 insects-13-00930-t009:** Chemical profiling of the aqueous extract of *A. halimus*.

RT	Exp. MS	Calc. MS	∆MS	MS/MS	Proposed Compounds	Ref.
1.05	191.02	191.02	0.0020	173, 129, 111	Citric acid	[31]
1.35	117.02	117.02	0.0006	99, 73	Succinic acid	[30]
1.39	161.05	161.05	0.0005	161, 143, 99	Hydroxymethylglutaric acid	
2.01	315.07	315.07	0.0000	315, 152	Protocatechuic acid, *O*-glucoside ether	[32]
2.71	359.10	359.10	0.0000	197, 153	Syringic acid *O*-glucoside ether	[33]
3.64	285.06	285.06	0.0000	285	Protocatechuic acid, arabinoside ether	
5.91	903.24	903.24	−0.0021	741, 314	Bidesmodic isorhamnetin, tetraglucoside ether	
6.40	557.06	557.06	−0.0011	477, 315	Isorhamnetin, glucoside monosulfate	
7.32	741.19	741.19	−0.0011	710, 314	Bidesmodic isorhamnetin, diarabinoglucoside ether	
7.40	769.22	769.22	−0.0017	769, 314	Bidesmodic Isorhamnetin, dirhamnoglucoside ether	
7.48	755.20	755.20	−0.0011	314	Atriplexoside A	[18]
7.64	609.15	609.15	−0.0009	609, 315	Isorhamnetin, glucoarabinoside ether	
7.69	639.16	639.16	0.0002	345	Spinacetin, arabinoglucoside ether	
7.75	525.31	525.31	−0.0004	479, 319, 159	20-Hydroxyecdysone or 22-epi-20-hydroxyecdysone	[27]
7.78	541.30	541.30	−0.0007	495	20-Hydroxyecdysone derivative	
7.87	623.16	623.16	−0.0008	623, 315	Azaleatin rutinoside or narcissin	[20]
8.19	395.01	395.01	−0.0001	315	Isorhamnetin sulfate	
8.19	507.11	507.11	−0.0004	507, 344	Spinacetin, glucoside ether	
8.55	733.11	733.11		653, 477, 315	Isorhamnetin, O-ferruloylglucoside, monosulfate	
8.58	825.43	825.43	−0.0016	825, 663, 487, 113	Olean-12-en-28-oic acid, dihydroxy-, glucoglucuronic acid ether	
8.59	971.48	971.49	−0.0010	810, 748, 629, 585, 539	Olean-12-endioic acid, diglucoside ether-, glucoside ester	
8.72	649.18	649.18	−0.0028	649, 563, 429, 315, 219	Dimorphamide A derivative	[28]
8.72	633.18	633.19	−0.0061	457, 439, 193, 175, 163	Dimorphamide A, mono acetyl derivative	[28]
8.78	449.14	449.14	0.003	449	Unknown	
8.94	823.41	823.41	−0.0014	823, 661, 555	Olean-12-en-28-oic acid, monohydroxy-, monooxo-, glucoglucuronic acid ether	
9.27	795.42	795.42	−0.0006	795, 675, 663	Olean-12-en-28-oic acid saponin	
9.35	823.41	823.41	−0.0014	823, 661, 485, 315, 241	Olean-12-endioic, monooxo-, diglucoside ether	
9.55	497.07	497.07	0.0005	497, 377, 215	Unknown	
9.82	853.42	853.42	−0.0013	854, 691,647, 629, 515	Olean-12-en-28-oic acid, dihydroxy, monooxo-, glucoglucuronic acid ether-, methyl ester	
9.95	645.18	645.19	−0.0033	645, 469, 425, 219, 193	Dimorphamide A derivative	[28]
10.02	807.42	807.42	−0.0012	807, 687, 645, 583	Olean-12-en-28-oic acid, monooxo-, glucoglucuronic acid ether	
10.08	825.43	825.43	−0.0015	825,770, 707, 663, 645	Olean-12-en-28-oate monohydroxy, monooxo, glucoside ether, glucoside ester	
10.23	809.43	809.43	−0.0014	809, 647, 585, 175	Olean-12-endioic acid, glucuronic acid ether-, glucoside ester	
10.27	851.41	851.41	−0.0009	851, 689, 627, 513, 113	Olean-12-endioic acid, glucoglucuronic acid ether-, methyl ester	
10.40	999.48	999.48	−0.0008	999, 838, 776, 657,613	Olean-12-endioic acid, diglucoside ether-, glucoside ester, methyl ester	
10.6	821.40	821.40	0.0000	821, 659, 597, 483, 175	Olean-12-en-28-epoxyoyl monocarboxylic, monooxo-, diglucoside ether-	
10.60	867.44	867.44	−0.0018	659	Olean-12-en-28-oic acid, monohydroxy, acetylglucoside ether, glucoside ester	
10.6	659.34	659.34	−0.0009	659, 583, 483, 175, 113	Olean-12-en-28-oic acid, dioxo, glucuronic acid ether	
10.83	969.47	969.47	−0.0003	969, 849, 807, 789, 763	Olean-12-en-28-oate dioxo, diglucoside ether, glucoside ester	
10.92	809.43	809.43	−0.0010	809, 629, 585, 539	Olean-12-endioic acid, monooxo, glucoside ether, glucoside ester	
11.03	837.43	837.43	−0.0020	837, 717, 675	Olean-12-en-28-oate monocarboxylic, glucoglucuronic, methyl ester	
11.05	849.43	849.43	−0.0018	849, 807, 687, 627, 583	Olean-12-en-28-oate dioxo, glucoside ether, acetylglucoside ester	
11.09	897.45	897.45	−0.0017	689	Olean-12-endioic acid saponin	
11.20	851.44	851.44	−0.0018	851, 809, 791, 689, 629	Olean-12-en-28-oate monohydroxyl, monooxo, glucoside ether, acetylglucoside ester	
11.35	955.49	955.49	−0.0016	955, 793, 731, 613, 569	Olean-12-en-28-oic acid, monooxo-, diglucoside ether, glucoside ester	
11.36	661.36	661.36	−0.0007	661, 585, 485, 113	Olean-12-en-28-oic acid, monohydroxy, monooxo-, glucuronic acid ether	
11.49	879.43	879.44	−0.0045	879, 837, 717, 657, 613	Olean-12-en-28-oate, monooxo, glucoside ether, acetylglucoside ester, methyl ester	
11.57	925.48	925.48	−0.0002	925, 763, 701, 613	Olean-12-en-28-oate, diglucoside ether, glucuronic acid ester	
11.64	689.35	689.35	−0.3543	689, 513, 113	Olean-12-en-28-oic acid, dioxo, glucuronic acid ether, methyl ester	
11.86	691.37	691.37	−0.0044	691, 515, 113	Olean-12-endioic acid, monohydroxyl, monooxo, glucuronic acid ether, methyl ester	
11.95	793.44	793.44	−0.0012	793, 631, 455, 175, 113	Olean-12-en-28-oic acid, glucoglucuronic acid ether	
12.07	679.37	679.37	−0.0005	679, 399, 369, 191	Olean-12-en-28-oic acid saponin	
12.29	853.46	853.46	−0.0009	645, 585	Olean-12-en-28-oic acid saponin	
12.51	835.45	835.45	−0.0009	835, 775, 673, 613, 569	Olean-12-en-28-oic acid, gluco(acetyl)glucuronic acid ether	
12.93	645.36	645.36	−0.0011	645, 469, 113	Olean-12-en-28-oic acid, monooxo, glucuronic acid ether	
13.12	807.42	807.42	−0.0013	807, 627, 583, 469, 99	Olean-12-en-28-oate, dioxo, glucoside ether, glucoside ester,	
13.35	675.37	675.38	−0.0033	675, 499, 113	Olean-12-en-28-oic acid, glucuronic acid ether, methyl ester	
13.47	647.38	647.38	−0.0005	647, 471, 113	Olean-12-en-28-oic acid, monohydroxy, glucuronic acid ether	
14.15	793.43	793.44	−0.0049	793, 613, 569, 523, 113	Olean-12-en-28-oic acid, glucuronic acid ether, glucoside ester	
14.57	687.38	687.38	0.0031	687, 645, 627, 583, 565	Olean-12-en-28-oic acid saponin	
15.05	631.38	631.38	0.0004	631, 455, 113	Olean-12-en-28-oic acid, glucuronic acid ether	

## Data Availability

All data are included within the manuscript.

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
