# Peer review of "Insecticidal Activities of Atriplex halimus L., Salvia rosmarinus Spenn. and Cuminum cyminum L. against Dactylopius opuntiae (Cockerell) under Laboratory and Greenhouse Conditions"

_insects, 2022, doi:10.3390/insects13100930_

Round 1

Reviewer 1 Report (Previous Reviewer 2)

#insects-1939151

 The authors have made corrections in the new version of the manuscript and provided some missing information in the first version. However, other shortcomings still need to be visited before manuscript publication.

 L 248-252: Does the number refers to individuals or colonies? In lines 250-252, the authors indicate that a scale of 1 to 2 refers to the number of colonies, while 3-6 to the percentual of coverage. Plus, this scale was not mentioned afterwards.

The number of colonies per replicate must also be included in the statistical model since the response was percentual. The different number of colonies per replicate can lead to over or underestimation, i.e., percentual mortality is colony-size dependent. For example, if in 20 colonies 10 died, it represents 50% of mortality. But the same percentual can be achieved if in 10 colonies 5 died.

L 317: Species name not italicized.

L 324-333: The authors refer to the estimated lethal concentrations as sublethal. But lines 335-337 they indicate that as lethal concentrations. Confidential intervals are not informed, which is essential information to verify if the concentrations are similar or not. Chi-square values are missing either in tables 4 and 5.

Tables 6 and 7: These tables can be merged to present consistent results. It’s not clear if they represent the same experiment or not. For instance, data of 3 d in table 6 are not similar to table 7 (first spray). 

Author Response

Reply to Reviewer 1

The authors have made corrections in the new version of the manuscript and provided some missing information in the first version. However, other shortcomings still need to be visited before manuscript publication.

L 248-252: Does the number refers to individuals or colonies? In lines 250-252, the authors indicate that a scale of 1 to 2 refers to the number of colonies, while 3-6 to the percentual of coverage. Plus, this scale was not mentioned afterwards.

Number refers to colonies number which represent medium level of infestation that ranged between 41 to 80 colonies per cladode. This range is depending on size of the cladode.

Small size of clade with 41 colonies is defined as medium infestation. Big size of cladode with up to 80 colonies is considered also as medium infestation. We took into consideration all factors for better ranking of infestation level. Sentence was reformulated as below:

Treatments were applied to selected cladodes at medium infestation levels (26-50%) (between 41 to 80 per cladode) using a modified rating scale from Silva (1991).

The number of colonies per replicate must also be included in the statistical model since the response was percentual. The different number of colonies per replicate can lead to over or underestimation, i.e., percentual mortality is colony-size dependent. For example, if in 20 colonies 10 died, it represents 50% of mortality. But the same percentual can be achieved if in 10 colonies 5 died.

The mortality of colonies cannot be determined, since each colony is composed from different adult females. It’s impossible to determine the mortality without evaluating each female. The evaluation takes time. The dead females revealed a desiccation of their bodies and dark brown color.

L 317: Species name not italicized.

Corrected

L 324-333: The authors refer to the estimated lethal concentrations as sublethal. But lines 335-337 they indicate that as lethal concentrations. Confidential intervals are not informed, which is essential information to verify if the concentrations are similar or not. Chi-square values are missing either in tables 4 and 5.

The word Sublethal was removed and replaced by lethal.

The estimated lethal concentrations were calculated manually in Microsoft Excel, by using the graph equation for each exposure time.

Tables 6 and 7: These tables can be merged to present consistent results. It’s not clear if they represent the same experiment or not. For instance, data of 3 d in table 6 are not similar to table 7 (first spray). 

The first table (6) represents the application of the treatment on D. opuntiae nymphs, while the second table (7) represents the result of the application on adult females D. opuntiae on days 3, 5 and 7 after the first and second application. Also, the time for their evaluation is different 7 days after firs application for female and 14 days after second spray. While, only 4 days after sprays is needed to evaluate the efficacy on nymphs.

Reviewer 2 Report (Previous Reviewer 1)

The article “Insecticidal activities of Atriplex halimus L., Salvia rosmarinus Spenn. and Cuminum cyminum L. against Dactylopius opuntiae (Cockerell) under laboratory and greenhouse conditions” has been improved, addressing the comments listed in the first review report.

The paper could be considered for publication after some minor revisions.

1)      Introduction: lines 44-45: although corrected, this sentence remain unclear

2)      Introduction, lines 130 and 132: Rosmarinus officinalis is synonym of Salvia rosmarinus, therefore the plant species should be cited using the valid name (S. rosmarinus); to avoid misunderstanding, in line 130 literature data could be reported as follows: “The major phytochemicals in S. rosmarinus (syn. R. officinalis) and …...”. Check and correct for the same reason also line 498.

3)      Lines 226-227: remove “We have sprayed 0.5 ml of the tested extract per petri dish.” the same sentence is reported at line 224

4)      Results: Statistical analyses tables: when the p-values are higher than 0.05, the letters indicating significant differences should be removed.

5)      Line 478: remove (OP) as not cited elsewhere in the manuscript

6)      References: something wrong in formatting all references (Capital letters for genera, authorities, states, or the first word in the article title, italic for species name, etc.), please check carefully (see file). Moreover, the papers cited as no. 19 and no. 27 are the same; please check all cited references and correct the paper numbers in the text when needed.

Kind regards

Author Response

Reply to Reviewer 2

The paper could be considered for publication after some minor revisions.

1)      Introduction: lines 44-45: although corrected, this sentence remain unclear

The sentence was improved

2)      Introduction, lines 130 and 132: Rosmarinus officinalis is synonym of Salvia rosmarinus, therefore the plant species should be cited using the valid name (S. rosmarinus); to avoid misunderstanding, in line 130 literature data could be reported as follows: “The major phytochemicals in S. rosmarinus (syn. R. officinalis) and …...”. Check and correct for the same reason also line 498.

Replaced

3)      Lines 226-227: remove “We have sprayed 0.5 ml of the tested extract per petri dish.” the same sentence is reported at line 224

The sentence was removed

4)      Results: Statistical analyses tables: when the p-values are higher than 0.05, the letters indicating significant differences should be removed.

The letters indicating non-significant was removed

5)      Line 478: remove (OP) as not cited elsewhere in the manuscript

Removed

6)      References: something wrong in formatting all references (Capital letters for genera, authorities, states, or the first word in the article title, italic for species name, etc.), please check carefully (see file). Moreover, the papers cited as no. 19 and no. 27 are the same; please check all cited references and correct the paper numbers in the text when needed.

Thank you, we have updated all the references and we have solved all the problems.

Round 2

Reviewer 1 Report (Previous Reviewer 2)

n/a

This manuscript is a resubmission of an earlier submission. The following is a list of the peer review reports and author responses from that submission.

Round 1

Reviewer 1 Report

The subject of the article “Insecticidal activities of Atriplex halimus L., Salvia rosmarinus Spenn. and Cuminum cyminum L. against Dactylopius opuntiae (Cockerell) under laboratory and greenhouse conditions” is interesting, but some major concerns need to be addressed before considering it for publication.

The paper should be improved and resubmitted for publication only after major revisions.

Main concerns:

1)      The number of treated and examined nymphs and females of D. opuntiae in the greenhouse tests is not reported either in M&M or in the Results

2)      Data analysis is conducted on values transformed for their normalization, but the Abbott’s formula should be also adopted to obtain the correct mortality, at least for the result of each test (the last day). This should be considered also when comparing the mortality obtained in the study with those from literature, to avoid incorrect interpretation of results.

3)      Statistical analyses (tables 1, 2, 3): is it correct to apply a test to compare the means if the p-values are higher than 0.05?

4)      In the greenhouse tests some different concentrations of the A. halimus acqueous and hydroalcoholic extracts have been used, but data analysis does not include the calculation of the LC50, which in this kind of studies allows comparing the results with most of literature, taking into account also the mortality obtained in the control

5)      In the greenhouse tests treatments were made on cladodes, please add something about the phytotoxicity (or lack of phytotoxicity)

6)      Introduction: add some literature information about the secondary metabolites of the 3 selected plant species.

When the Latin names of plants, animals, fungi, etc. are cited for the first time in the text, they must include Authority, please check this aspect along all the manuscript.

The point by point comments or suggestions are reported below.

Title: check the font size

Key words: halimus should be italic

Lines 43, 45: Cactus pear or prickly pear? Use always the same common name thorough the manuscript

Lines 44-45: “as a result of drought” what do you mean? Unclear

Line 47: “effectively closes the limiting vegetation” what do you mean? Unclear

Line 60: remove “, Opuntia ficus-indica (L.)”

Lines 60-61: “This wild cochineal was first detected in Sidi Bennour area, 70 km from El Jadida (western central Morocco) in September 2014”- please cite reference

Lines 71-72: sentence very similar to Ramdani et al., 2020

Line 74: ONSSA 2019 cite it here as reference (with a number) and add in Reference paragraph

Line 75: In addition to the resistance problems, ….

Line 82: “Different botanical extracts are used as eco-friendly insecticides to control and manage different crop pests and plant diseases [4] ……” This is a general sentence, the cited paper is about D. opuntiae, it could be removed considering that the paper 7 cited in the following line is a review on Biopesticides.

Lines 86-87: please correct the species names (Chenopodium ambrosioides and not Ambrosioides, etc.)

Lines 88, 91, 92 : add Authority for Capsicum annuum, Fusarium catingaense and Nicotiana tabacum

Lines 94-95: Atriplex halimus L., Salvia rosmarinus Spenn. and Cuminum cyminum L.

Line 104: S. rosmarinus….C. cyminum

Line 179: remove “(CRD)” as not cited elsewhere in the manuscript

Line193: please add more details about the rating scale used for the infestation levels of cladodes, and add (here or in Results) the number of treated insects (nymphs and females)

Lines 198-199: please explain more clearly the timing for the extract application to the insect females.

Lines 201-206: Statistical analyses: please see above (main concerns)

Line 207, 231, 242, 261, 268: please add also the DF for number of treated insects

Lines 329-344 move to discussion

Line 342: Atriplex farinosa Forssk

Line 344: Atriplex canescens, add Authority

Lines 351-362: These sentences summarize M&M and results, please rewrite leaving only information essential to a better comprehension of the following part.

Lines 367-368: The observed activities might be attributed to the high contents of saponins, which in A. halimus aqueous extract amounted to 24.09 mg…

Line 371: add references or add (unpubl. data)

Lines 376-378: El-Gougary (1998) has demonstrated the insecticidal activity of the ethanol extract of A. halimus against the cotton caterpillar Spodoptera littoralis (Lepidoptera: Noctuidae) with an LD50 value of 5.6 mg/larva. Add [32] or change the sentence in order to add the reference citation by number. Add Authority to Spodoptera littoralis species name.

Lines 379-383 this sentence is unclear, how many components have been used on T. castaneum (ethanolic extract of A. halimus, petroleum ether, insecticides at which %)? please rephrase. Moreover: Line 380: what is OP?; Line 382: adults of Tribolium castaneum - please add Authority

Lines 385, 386, 387, 390, 399: Atriplex laciniata, Heterotermes indicola, Rhyzopertha dominica, Monomorium pharaonic, Virginia tobacco - please add Authority

Line 390: remove (Pharaoh ant), redundant

Line 402: add reference (may be [34]?)

Line 406: However, a recent study….. - move [35] at the end of this sentence. Moreover, please add P. operculella Authority

Lines 411 and 413-14: Libidibia ferrea and Agave sisalana - please add Authority

Lines 418-420: these species have been already cited, please abbreviate the genus name and remove Authorities

Line 427: add reference number

Line 428: change lower with less (lower mortality)

Line 430: was and not were (insecticidal effect was enhanced)

Lines 431-432: please rephrase.

Lines 432-435: this sentence should be added in the introduction or in M&M

Line 437: its instead of their

Line 438: impact on natural enemies

Line 447: sensitive to the aqueous extract….

Lines 455-456: most effective for wild cochineal control.

Kind regards

Reviewer 2 Report

#insects-1858392

Naboulsi et al report the insecticidal activity of three plant species on nymphs and adults of the wild cochineal. The main authors’ findings indicate that Atriplex halimus is a potential source of new bioinsecticides to control that pest. However, the material and methods lack details and the results reported seem overestimated.

For instance, the volume of extract applied per replicate was not informed, either which consisted of a replicate in both experiments. The data was analyzed considering that each hour of the evaluation consisted of an independent experiment. It was not the case once the authors took evaluations on the same replicate through time. So, we expected data analysis as an ANOVA with repeated measures.

In addition, the manuscript is not in the journal guidelines (subsections indication and the simple summary are absent). There are other flaws, pointed out below, that should be addressed before the manuscript publication.

L 20: Simple summary is absent. Please check journal author guidelines.

L 25-26: Please consider removing ‘from in-region medicinal plants’ since the authors have already informed plant species names. It turns the statement unclear.

L 28: The mortality rates informed did not consider the mortality in control measures – Abbott formula. Please revise.

L 37: The safety was not tested here.

L 60: The Plant species was presented in complete form in line 43.

L 175 and L 177-1778: Since the authors tested 5% (line 175), why did they repeat the same concentration in further experiments (lines 177-178)?

L 177-179: What was the volume applied per plant? What consisted of each replicate – a plant?

L 191: How many individuals were transferred per plant?

L 194-195: Again, the volume applied per replicate was not informed and what consisted of each replicate either. Please provide this information to be precise. L 201-206: The authors need to analyze data considering repeated measures. In the current form, it indicates that for each hour of evaluation they performed independent experiments. It is not the case here. In addition, mortality values in treatments should be corrected by those noted in control using the Abbott formula to obtain effective mortality estimation for treatments. This absence leads to higher mortality rates for treatments, but control mortalities were also higher (=20%, see table 3; =41.25%, see table 4). Finally, Did the authors check ANOVA assumptions? If so, need to be informed here.

Tables 1, 2, 3, 4, and 5: Please inform standard deviation values for each mean informed in the tables. This information is necessary to show the mean deviation.

Table 5: The twice spray methodology was not described in the material and methods.

Discussion

The authors only argue their findings bringing information from other studies, and how the current study agrees or not with these previous findings. But other findings are not explained.

For instance:

(i) why did the authors evaluate mortality through time? It is due to the mode of action, we assume, which was not discussed here;

(ii) since the authors identified potential compounds, we expected that they present arguments to support this initial hypothesis of the main compounds responsible for the insecticidal activity. It may support future studies in prospecting compounds such desirable activity.

L 351-371: The paragraph can be reduced to avoid wordiness. The authors can consistently report the main findings without repeating material and methods.

L 437-438: which ones?

L 444-456: The authors need to revise the conclusion after the shortcomings mentioned in the data analysis.

L 467: References need to be revised with caution. Scientific names are not italicized.